# Recent Advancements in Metal and Non-Metal Mixed-Doped Carbon Quantum Dots: Synthesis and Emerging Potential Applications

**DOI:** 10.3390/nano13162336

**Published:** 2023-08-14

**Authors:** Zubair Akram, Ali Raza, Muhammad Mehdi, Anam Arshad, Xiling Deng, Shiguo Sun

**Affiliations:** 1Key Laboratory of Xinjiang Phytomedicine Resource and Utilization, Ministry of Education, School of Chemistry and Chemical Engineering, Shihezi University, Shihezi 832003, China; zubair.a.sidhu@gmail.com (Z.A.); razaali2021@stu.shzu.edu.cn (A.R.); anamarshad9293@gmail.com (A.A.); dxl_pha@shzu.edu.cn (X.D.); 2College of Chemistry & Pharmacy, Northwest A&F University, Xianyang 712100, China; kh.mehdiali@nwafu.edu.cn; 3College of Chemistry and Pharmaceutical Engineering, Hebei University of Science and Technology, Shijiazhuang 050018, China

**Keywords:** mixed-doped CQDs, chemical sensing, biosensing, synthesis techniques

## Abstract

In nanotechnology, the synthesis of carbon quantum dots (CQDs) by mixed doping with metals and non-metals has emerged as an appealing path of investigation. This review offers comprehensive insights into the synthesis, properties, and emerging applications of mixed-doped CQDs, underlining their potential for revolutionary advancements in chemical sensing, biosensing, bioimaging, and, thereby, contributing to advancements in diagnostics, therapeutics, and the under standing of complex biological processes. This synergistic combination enhances their sensitivity and selectivity towards specific chemical analytes. The resulting CQDs exhibit remarkable fluorescence properties that can be involved in precise chemical sensing applications. These metal-modified CQDs show their ability in the selective and sensitive detection from Hg to Fe and Mn ions. By influencing their exceptional fluorescence properties, they enable precise detection and monitoring of biomolecules, such as uric acid, cholesterol, and many antibiotics. Moreover, when it comes to bioimaging, these doped CQDs show unique behavior towards detecting cell lines. Their ability to emit light across a wide spectrum enables high-resolution imaging with minimal background noise. We uncover their potential in visualizing different cancer cell lines, offering valuable insights into cancer research and diagnostics. In conclusion, the synthesis of mixed-doped CQDs opens the way for revolutionary advancements in chemical sensing, biosensing, and bioimaging. As we investigate deeper into this field, we unlock new possibilities for diagnostics, therapeutics, and understanding complex biological processes.

## 1. Introduction

Carbon quantum dots (CQDs) are a fascinating class of fluorescent nanomaterials with particle sizes of between 2–10 nm [1]. Carbon quantum dots may also be referred to as carbon dots (CDs) [2], carbon nanodots (CNDs) [3], carbon polymer dots [4], and carbon nanoparticles (CNPs) [5]. In 2004, Xu et al. discovered these nanoscale structures coincidently during the isolation of single-walled carbon nanotubes synthesized from arc-discharge dust [6]. Since then, CQDs have received considerable attention owing to their distinctive properties and emerging applications.

Lately, CQDs have been synthesized for modification, surface functionalization, and various other applications [7]. However, many review articles published since 2011 have focused on the advancements in synthesizing carbon quantum dots using economical and straightforward methods [8], primarily for their applications in chemical/biological sensing and bio-imaging [9]. Hence, there is an urgent need for developing CQDs (particularly doped/co-doped) to further explore their applications [10].

## 2. Classification of CQDs

An essential characteristic of CQDs is their ability to be co-doped with metallic and nonmetallic elements or both [10,11]. Co-doping is a deliberate introduction of various elements into the carbon matrix which can improve the photoluminescence [12,13], optical [14], and catalytic properties of CQDs [15]. By incorporating metal and non-metal dopants, it is possible to fine-tune the optical and electronic properties of CQDs, thereby opening up new avenues for various applications [16].

### 2.1. Undoped Carbon Quantum Dots (CQDs)

In un-doped CQDs, carbon atoms exit in sp^2^ and sp^3^ hybridization in the presence of hydrogen (H) and oxygen (O) atoms [17]. Fruit and vegetable extracts, rice flour, banana plant, cashew gum, eggs, hair fibers, almond, honey resins, PEG, citric acid, ascorbic acid, folic acid, glycerol, and many others are their naturally occurring sources [12,18,19,20].

### 2.2. Doped/Co-Doped CQDs

Doped or co-doped CQDs are mainly synthesized by doping hetero-atoms such as boron (B), fluorine (F), phosphorus (P), sulphur (S), and nitrogen (N) in the general composition. Doping and co-doping regulate the photoluminescence phenomenon in CQDs to enhance their fluorescence intensity, calculated as quantum yield (QY%) [21,22].

### 2.3. Mixed-Doped CQDs

Mixed-doped CQDs are prepared by doping CQDs with metals like zinc (Zn), iron (Fe), cobalt (Co), nickel (Ni), chromium (Cr), and non-metals like N, B, and P, which leads to the development of novel materials with enhanced photoluminescence. Mixed-doped CQDs show the properties like fast response time, pH compatibility, excellent detection limit, high sensitivity, and selectivity [23].

Current developments in the preparation and applications of doped and co-doped CQDs need to be sufficiently evaluated, despite further advancements in CQDs. Most existing review articles [7,9] focus primarily on the doping or co-doping of CQDs, whereas other emerging areas still need to be explored.

This review is focused on the latest developments in producing mixed-doped carbon quantum dots (CQDs) and their diverse potential applications, as illustrated in Figure 1. These applications include chemical or bio-sensing, bio-imaging, optical sensors for detecting metal ions, drugs, gene delivery, and temperature sensing.

## 3. Synthesis Techniques of Mixed-Doped Carbon Quantum Dots

### 3.1. Synthesis of Fe-N-CQDs

The electrochemical oxidation method was adopted to synthesize Fe-N-CQDs by coating an electrode with carbon cloth. Fe-N-C and PTFE in the ethanolic mixture were coated on the carbon fabric to prepare an electrode. The electrode worked as an anode and was dried at 80 °C for 12 h for electrolysis. PTFE material and voltage conditions were controlled to assess its efficiency. The counter electrode was Pt foil with 8 mL H_2_O, 30 mg NaOH, and 35 mL C_2_H_5_OH as an electrolyte which was further dialyzed for 24 h, and the final solution was lyophilized to obtain Fe-N-CQDs powder (Figure 2a) [23].

### 3.2. Synthesis of Fe-N-CNPs

The present study involved the production of a composite material, namely, Fe-N-CNPs, through a plasma process. The plasma was generated by applying a high-voltage pulse between a proportional pair of metallic Fe electrodes immersed in a homogeneous blended CNF and 2-cyanopyridine (C_4_H_4_N_2_). The carbon nanoparticles were doped by nitrogen in situ and were synthesized through dissociation and recombination processes using 2-cyanopyridine as the precursor. Simultaneously, iron atoms were mixed from the electrode into the homogeneous mixture. Within the plasma region, Fe atoms may react with C_2_ and CN radicals that possess high levels of energy, which are produced through the dissociation of 2-cyanopyridine molecules. This reaction results in the creation of Fe-N coordination. The Fe and N sites in the plasma region were intermingled into suspension, and, subsequently, adhered to the surface of the CNF, forming a composite structure comprising N-CNP. The synthesized substance underwent a second heat treatment at 900 °C for 2 h within an argon atmosphere (Figure 2e) [24].

### 3.3. Synthesis of the Fe,N-C Electrocatalysts

In this synthesis process, 0.4 g Hemin was initially dissolved in 40 mL N,N-dimethylformamide (DMF), and combined with 0.4 g BP2000. The resulting mixture underwent sonication for a duration of 1.5 h to produce a uniform solution, which was, subsequently, subjected to a temperature of 50 °C to facilitate solvent evaporation. Later, the powder was transferred into a tubular furnace for pyrolysis at 900 °C for 2 h while exposed to an argon atmosphere. The resultant substance was washed using 2 M HCl at ambient temperature to eliminate unstable iron atoms or compounds. After washing with the acid, the substance underwent a second heat treatment at a temperature of 900 °C for 2 h in an argon atmosphere, producing a modified Fe-N carbon electrocatalyst, Fe-N-C-AH (Figure 2f) [25].

### 3.4. Synthesis of Pristine Fe-N-GQDs

A solution was prepared by dissolving 0.25 g citric acid, 0.16 g urea, and 0.2 g FeCl_3_ in 5 mL ultrapure deionized water. After an hour of stirring, the solution was transferred into a 100 mL autoclave and was heated up to 150 ℃ for 12 h. Then, the mixture was centrifuged at a speed of 10,000 RPM for 5 m to eliminate solid residue. The aqueous solution underwent dialysis using a dialysis membrane for one night. The Fe-N-GQDs were obtained as the outcome [26].

### 3.5. Synthesis of Fe/N-CDs

The solution was prepared by dissolving 0.06 g FeCl_3_·6H_2_O, 0.10 g EDTA, and 0.06 mL DETA in 10 mL ultrapure water. Then, the mixture was introduced into an autoclave lined with 30 mL Teflon, which was then subjected to a temperature of 200 °C for 6 h. The resultant solution was subjected to a 2500-fold dilution using PB buffer (10 mM, pH = 7.4) and, subsequently, preserved at 4 °C for further use (Figure 2b) [27,28].

### 3.6. Synthesis of Fe,N-CDs

The experimental procedure involved the combination of 1 g trisodium citrate, 2 mL ethylenediamine, and 1 g FeCl_3_ with 30 mL ethylene glycol. The solution underwent vigorous stirring for 15 m until achieving clarity. Subsequently, the mixture was introduced into a stainless-steel autoclave lined with 100 mL Teflon. Then, the solution was subjected to a temperature of 200 °C and maintained for 8 h. After cooling at an ambient temperature, the products with a dark-brown hue underwent centrifugation at 10,000 RPM for 10 m to eliminate the least dense particles. Later, the rotary evaporator was employed to eliminate ethylene glycol. Fe,N-CDs in solid form were dissolved into deionized water and subjected to dialysis to eliminate low molecular weight species. Afterwards, the Fe,N-CDs solution was lyophilized to obtain its powdered form. The powder was stored at a temperature of 4 °C to ensure its preservation for later use (Figure 2c) [29].

### 3.7. Synthesis of Fe@N-CDs

The Fe@N-CDs were synthesized using *P. edulis* as an eco-friendly carbon precursor in a single-step hydrothermal method. The husks of *P. edulis* were first subjected to natural drying and then pulverized into a fine powder. After that, a mixture comprising 1.0 g powder, 1.0 mL EDA (agent for surface passivation), and 0.6 g Fe_2_(SO_4_)_3_ as an iron source was homogeneously blended with 35 mL distilled water. Later, the solution was introduced into an autoclave lined with 50 mL Teflon and subjected to thermal treatment at 180 °C for 4 h. After the reaction, the autoclave was allowed to undergo natural cooling to achieve ambient temperature. The resulting carbonaceous solution was filtered using a 0.22 μm membrane filter to remove particles of significant size. Afterward, the solution was purified using a dialysis bag with deionized water for 24 h. A solution of Fe@NCDs exhibiting a homogenous brown hue was successfully obtained and preserved at a temperature of 4 °C for further use (Figure 2d) [30].

**Figure 2 nanomaterials-13-02336-f002:**
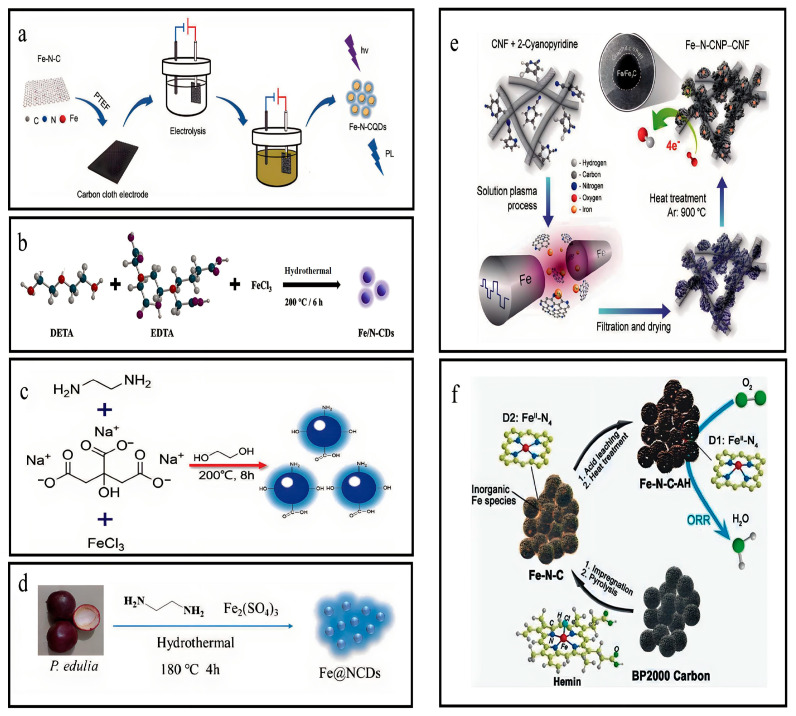
Overview of synthesis of iron, nitrogen co-doped CQDs. (**a**) Synthesis of Fe-N-CQDs. Reproduced from [23], with permission from Elsevier 2021. (**b**) Synthesis of Fe/N-CQDs. Reproduced from [27,28], with permission from Elsevier 2022. (**c**) Synthesis of Fe,N-CDs. Reproduced from [29], with authorization from RSC 2021. (**d**) Synthesis of Fe@NCDs. reproduced from [30], with permission from Springer 2020. (**e**) Synthesis of Fe-N-CNP. reproduced from [24], with permission from (**f**) Synthesis of Fe-N-C. Reproduced from [25], with permission from RSC 2017.

### 3.8. Synthesis of N,Cu-CQDs

In a standard protocol, a solution containing 19 mmol of EDTA and 19 mmol of Cu(NO_3_)_2_·3H_2_O was combined with 105 mL of ultrapure water within a 150 mL Teflon-lined autoclave. The resultant mixture was subjected to a thermal treatment at a temperature of 180 °C for 12 h after ultrasonic irradiation for 15 min. After achieving the ambient temperature, the solution mixture was subjected to overnight dialysis using a bag with a molecular weight retention threshold 1000 Da, for the complete removal of salt ions. The solution was later filtered using a 0.22 μm filter to eliminate particles of significant size. Then, the solution was subjected to re-dissolution using ultrapure water until a total volume of 100 mL was achieved following lyophilization. The N,Cu-CD concentration was measured at 0.73 mg mL^−1^ (Figure 3b) [31].

### 3.9. Synthesis of Cu-N@CDs

The Cu-N@CDs were synthesized by the thermolysis technique. A mixture of 1 g of Na_2_-EDTA and CuCl_2_ mixture was introduced into a 50 mL beaker containing deionized water. Then, the solution was stirred mechanically for 20 m. The product was isolated via filtration and heated in an oven at 200 °C for 5 h. Subsequently, the oven was allowed to cool at RTP, and the resulting product was collected, pulverized, and dissolved in deionized water. The suspension underwent ultrasonic treatment for 2 h at an ambient temperature. The filtration membrane (with a pore size of 0.22 μm) was utilized to eliminate the sizable particles via filtration. The acquired Cu-N@C-dots solution was preserved at a temperature of 4 °C to facilitate subsequent analysis and utilization in PGL estimation (Figure 3e) [32].

### 3.10. Synthesis of N/CuCDs

The N/CuCDs were synthesized by a one-pot hydrothermal approach. A solution was prepared by adding 1.2 g citric acid monohydrate (CA) and 0.15 g copper acetate monohydrate in 20 mL of deionized water. Diethylenetriamine (DETA) was dissolved in the resultant solution at 0.15 mL. The solution underwent ultrasonic dissolution for 10 min, and, then, it was introduced into a Teflon-lined stainless-steel autoclave with a volume capacity of 30 mL. The autoclave was subjected to a temperature of 230 °C for 12 h. The product was dialyzed for 72 h after cooling at an ambient temperature to eliminate residual reactants. The hazel solution obtained was subjected to reduced pressure distillation to eliminate excess water. The N/CuCDs brown powder was acquired through vacuum drying (Figure 3d) [33].

### 3.11. Synthesis of Cu-N-CDs

The Cu-N-CDs were synthesized by solvothermal carbonization of folic acid (FA) and CuCl_2_ in an ethanolic solution. A solution was prepared by dissolving 0.05 g FA and 0.12 g CuCl_2_ in 15 mL ethanol. The solution was introduced into an autoclave lined with polytetrafluoroethylene and heated up to 180 °C for 6 h. The unrefined substance underwent filtration using a nylon membrane filter with a pore size of 0.22 μm and then eliminating ethanol through rotary evaporation. The obtained product was dialyzed in water for approximately 24, 36, and 48 h by adding deionized water. The findings suggest that variations in dialysis time do not have a significant impact on the fluorescence intensity. Consequently, a dialysis time of 24 h was the most effective. The obtained CDs are commonly denoted as Cu-N-CDs (Figure 3a) [34].

**Figure 3 nanomaterials-13-02336-f003:**
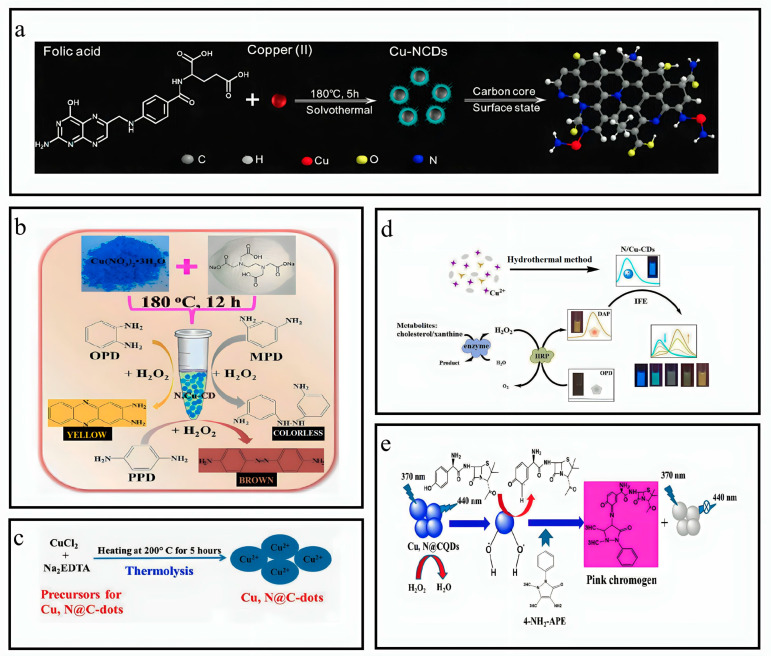
Overview of synthesis of copper and nitrogen co-doped CQDs. (**a**) Synthesis of Cu-N-CDs. Reproduced from [34], with permission from Elsevier 2020. (**b**) Synthesis of N,Cu-CD. Reproduced from [31], with permission from Springer 2019. (**c**) Synthesis of Cu,N@C-dots. Reproduced from [35], with permission from Springer 2022. (**d**) Synthesis of N/Cu-CDs. Reproduced from [33], with permission from ACS 2017. (**e**) Synthesis of Cu,N@CQDs. Reproduced from [32], with permission from Springer 2019.

### 3.12. Synthesis of Cu,N@CQDs

The Cu, N@CQDs were synthesized using a one-pot hydrothermal approach. In summary, a solution was prepared by dissolving 0.65 g glucose and 0.45 g CuSO_4_·5H_2_O in 50 mL of double-distilled water. A 0.35 mL of ethylenediamine was introduced into the mixture above. Later, the mixture was agitated for 20 min and then followed by the thermal treatment at 200 °C for 10 h. The solution was obtained by adding the sample into centrifugation at 6000 RPM for 15 min. The resulting solution was filtered using a 0.2 μm filter membrane and was subjected to dialysis within a dialysis bag for three days. The dialysis water was replaced every 12 h. The synthesized Cu, N@CQDs were freeze-dried and stored at a temperature of 4 °C (Figure 3c) [35].

### 3.13. Synthesis of Zn-N-CQDs

A solution was prepared by dissolving 0.5 g ZnSO_4_, 0.9 mL of conc. H_2_SO_4_, and 2.65 mL of ethylene diamine in 25 mL of water. The mixture was subjected to ultrasound for 5 min to ensure complete homogenization. The experiment was conducted for 20 min at a temperature of 150 °C, using a microwave digestion method with a power output of 1450 W. After removal, the substance was cooled to room temperature, gathered, and concentrated for further use. The QY% of Zn-N-CQDs was determined to be 14.26% using the relevant equation (Figure 4a) [36,37].

### 3.14. Synthesis of N,Zn-CDs

The synthesis process involved the addition of 0.9 g of citric acid, 2.6 mL EDA, and 0.2 g zinc acetate to 25 mL of distilled water. The resulting mixture was subjected to sonication for a brief period until a clear solution was obtained. The transparent solution was placed in a Teflon vessel with a volume of 60 mL, which was sealed tightly and inserted into a microwave oven. The microwave oven, operating at 850 W, raised the system’s temperature to 150 °C within a minute. Under the specified reaction conditions, the reaction was conducted for 30 min by regulating the exposure time of microwaves (700 W). After 30 min, the reaction setup was fully deactivated and then allowed to equilibrate at an ambient temperature. The reaction solution was centrifuged at 10,000 revolutions per minute for 15 min at room temperature. Afterward, filtration was carried out using a membrane of 0.22 μm, followed by dialysis through a membrane for 8 h, using ultrapure water. The solid materials were acquired for subsequent analysis through freeze-drying (Figure 4b) [38].

### 3.15. Synthesis of Zn/Co-N-CQDs

A mixture of 0.3 g diphenyl semi-carbazide, 0.1 g cobalt sulfate, and 0.1 g zinc chloride was prepared in 30 mL aqueous solution. The solution mixture was transferred into a polytetrafluoroethylene auto-clave and subjected to a thermal treatment at 200 °C for 12 h. After cooling to room temperature, the solution was dialyzed through ultrapure water for 24 h, using a dialysis membrane with a molecular weight cut-off of 1000 Da. The ultrapure water used in the dialysis process was replaced every 4 h. Subsequently, the Zn/Co-NCDs solution was acquired and preserved at 4 °C for further examination [39].

### 3.16. Synthesis of Ce-N-CQDs

A solution in 40 mL ultrapure water was prepared by dissolving 2 g citric acid (CA) and 1 g cerium nitrate hexahydrate (Ce(NO_3_)_3_·6H_2_O). Later, 1 mL ethylenediamine (EDA) was added to the prepared solution and stirred continuously to achieve homogeneity. Following this, the mixture underwent a reaction within an autoclave at a temperature of 220 °C for 2 h. After cooling to room temperature, the brown–yellow substance was purified through dialysis. Subsequently, the Ce-N-CQDs powder acquired through lyophilization was enclosed and conserved at a temperature of 4 °C for further investigations (Figure 5a) [40,41].

### 3.17. Synthesis of Ni-N-C Materials

To prepare Ni-N-C materials, PVP and Ni(NO_3_)_2_ were completely dissolved into a concentrated solution of NaCl. The resulting mixture was, subsequently, frozen at −80 °C, and water was removed gradually from the frozen mixture through vacuum freeze drying. Then, the freeze-dried powder was pyrolyzed to obtain Ni-N-C materials under an N_2_ atmosphere at a temperature of 900 °C (Figure 5b) [42].

### 3.18. Synthesis of Au/N-CQDs

The Au/N-CQDs were synthesized via hydrothermal technique. Initially, a solution comprising 0.1 g folic acid (FA), 1 mL glycerol, and 1 mL chloroauric acid (CA, HAuCl_4_·3H_2_O) was dissolved in 30 mL deionized water. Then, the solutions were subjected to magnetic heating agitation at a temperature of 80 °C for 10 min. The solutions were introduced into a 50 mL autoclave lined with poly(tetrafluoroethylene) (Teflon) and subjected to a temperature of 180 °C for 12 h. Finally, the autoclave was allowed to cool at room temperature, resulting in formation of yellowish solutions. The solutions were filtered to eliminate aggregate using microfiltration membranes with a pore size of 0.22 μm. Finally, solutions of Au/N-CQDs were acquired (Figure 5d) [44].

### 3.19. Synthesis of Co-N-CDs

The conditions that yielded the most favorable outcome for the synthesis of Co-N-CDs were determined to be as follows: The experimental conditions involved the use of 2.0 g CA, 0.07 mL DETA, and 0.4 g CoCl_2_·6H_2_O, with a reaction temperature of 160 °C and a reaction time of 8 h. A solution was prepared to synthesize Co-N-CDs by dissolving 2.0 g CA and 0.4 g CoCl_2_·6H_2_O in 20 mL ultrapure water. After that, 0.07 mL DETA was introduced into the solution mentioned above. The solute underwent complete dissolution by employing an ultrasonic technique for 10 min. Then, the solution was introduced into a Teflon-lined stainless-steel autoclave chamber with a 50 mL volume and subjected to a reaction at 160 °C for 8 h. After cooling to room temperature, the solution was subjected to dialysis using a dialysis bag with a molecular weight cutoff of 500 Da for 72 h. The Co-N-CDs solid powder was acquired through a 48 h vacuum freeze-drying process. The Co-N-CD powder was stored at a temperature of 4 °C for further use. (Figure 5c) [43,45].

### 3.20. Synthesis of Bi-N-CQDs

DI water was used to wash the collected rice husk to remove the significant contaminants in the rice husks. Then, the rice husk was processed via a tabletop blender into tiny fragments. After pretreatment with 0.1 M HCl to eliminate any surface contaminants, the fine rice husk was continuously rinsed with DI water until the pH reached neutral. Then, the rice husk was dried at 60 °C in an oven for the rest of the day. For the doping of N and Bi, EDA with a volume ratio of 5 and 20 vol% and bismuth nitrate pentahydrate with a mass of 1 and 5 wt% were used. Following the various academic research for the synthesis of carbon quantum dots from rice husk, the hydrothermal temperature of 190 °C was chosen for synthesizing Bi-N-CQDs [46].

### 3.21. Synthesis of Mg-N-CQDs

Typically, 40 mL of distilled water was used to dissolve 10 g of citric acid (CA) and 4.2 g of Mg(OH)_2_ to prepare a colorless, transparent, homogenous solution to avoid other anions. The hydrothermal method was used at a temperature of 200 °C for 3 h. Later, the product was subjected to filtration, dialysis, and lyophilization to obtain CQDs labeled as Mg-CQDs. Corresponding experiments were conducted to understand the role of magnesium chelator for high yield fluorescent yield of carbon quantum dots (CQDs), such as (1) citric acid pyrolysis, (2) use of ethylenediamine as surface modification agent in the absence of Mg(OH)_2_, and (3) incorporation of ethylenediamine and Mg(OH)_2_. The resulting product was labeled as Mg-N-CQDs (Figure 6a,b) [47,48,49].

### 3.22. Synthesis of Ag-N-CQDs

The Ag-N-CQDs synthesis involved mixing 1 mL of EDA, 0.1 g AgNO_3_, and 1 g citric acid (CA). Then, the solution mixture was transferred to a Teflon-coated container at a temperature of 180 °C for 4 h. The mixture was dialyzed for 24 h to obtain Ag-N-CQDs by cooling to room temperature [50,51,52].

### 3.23. Synthesis of N/Al-CDs

The synthesis of N/Al-CDs involved the combination of finely ground durian shell waste (DSW) powder weighing 1 g and having a mesh size of 0.22 mm, with 0.1 g urea and 0.1 g aluminum nitrate. Later, the mixture was introduced into a Teflon-coated stainless-steel autoclave reactor with a 45 mL volume and 25 mL of deionized water. Then, the sealed reactor was subjected to a thermal treatment at a temperature of 210 °C for 12 h once the mixture had reached ambient temperature [53].

### 3.24. Synthesis of Zr-N-CDs

Zr-N-CDs were produced by dissolving 2 g citric acid and 1 g zirconium chloride in 40 mL ultrapure water, later supplemented with 1 mL ethylenediamine. After complete dissolution in water, the two mixtures above were transferred to a lined autoclave and subjected to a temperature of 150 °C for 2 h. The following procedure was conducted using identical parameters. Following the cooling of the mixture to ambient temperature, it underwent centrifugation for a duration of 15 m at a velocity of 10,000 revolutions per minute. The resultant supernatant was collected and subjected to additional filtration via a 0.22 mm filtering apparatus, yielding a homogeneously dispersed brown solution. The consistent particle size and a high quantum yield of the fluorescent CDs were ensured by transferring the solution to an interception bag with an interception quantity of 3000 Da. The interception bag underwent a 24 h purification process in ultrapure water at ambient temperature. Afterward, the solution was refrigerated for 12 h, followed by drying the samples through a freezer dryer, producing solids that exhibit commendable water solubility [54].

## 4. Applications of Mixed-Doped CQDs

### 4.1. Applications of Mixed-Doped CQDs in Chemical Sensing

#### 4.1.1. Sensitivity of Fe^3+^

This study demonstrated the effective detection of Fe^3+^ ions ranging from 0.05 μM to 125 μM using N-Zn-CDs, resulting in a significant change in N-Zn-CDs’ fluorescence intensity. The calibration equation for this relationship is expressed as F/F_0_, exhibiting a coefficient (R^2^) value of 0.9983 (Figure 7b). In this context, the variables F and F_0_ denote the fluorescence emission intensity of N-Zn-CDs when Fe^3+^ is present and absent, respectively. As per the Environmental Protection Agency (EPA) rule, the detection limit in drinking water was significantly lower than the maximum allowable level [55,56]. The study’s findings indicated that the synthesized N-Zn-CDs exhibited a notable sensitivity towards Fe^3+^ ions. These results contribute to expanding research on fluorescence sensors that detect Fe^3+^ ions. The result of the inner filter effect (IFE) is reinforced by the overlap of the absorption spectrum of the analyte or quencher with the excitation or emission spectrum of N-Zn-CDs [57,58]. This study aimed to examine the selective sensing capabilities of N-Zn-CDs in the presence of different metal ions in an aqueous solution. The quenching in fluorescence intensity of N-Zn-CDs was not significant for all metals except Fe^3+^. Many functional groups on the N-Zn-CDs surface may result in a significant complex formation with Fe^3+^ ions. Moreover, it has been observed that Fe^3+^ ions exhibit a higher affinity towards functional groups enriched with oxygen compared to other metal ions. The N, Zn-CDs demonstrated a significant response to Fe^3+^ across a concentration range of 0.05–125 μM. Furthermore, the feasibility of utilizing the synthesized N, Zn-CDs as a sensing mechanism for detecting Fe^3+^ ions was assessed through experiments conducted on water samples obtained from the river and circulating water. Therefore, this study presents a synthesis method that can be readily adjusted for use in various environmental applications [38,59,60].

#### 4.1.2. Determination of Mn(VII)

N/Al-CDs’ high-water solubility and optical stability were utilized to create a fluorescent nanoprobe capable of detecting dissolved Mn^7+^ in water with sensitivity and selectivity. This research explored the selectivity of N/Al-CDs for detecting Mn^7+^ by analyzing the fluorescence changes in the presence of various metal ions (Figure 7c) [53]. The fluorescence intensity of N/Al-CDs was reduced to 85% upon adding Mn^7+^. At the same time, the presence of other metals, except Fe^3+^ and Cr^6+^, only caused a minor change in intensity of approximately ±5%.

Nevertheless, incorporating the N-Al doping mixture has resulted in a notable decrease in the quenching phenomenon, amounting to a reduction of 20%. This reduction has been recognized as forming Al-O bonds, which hinder the formation of Fe^3+^ ground-state complexes with oxygen groups. Therefore, N/Al-CDs exhibited enhanced selectivity towards the Mn^7+^. The N/Al-CDs synthesized in this study demonstrate potential as a highly effective fluorescent sensor for detecting Mn^7+^. The synthesized N/Al-CDs, which exhibit high selectivity and a limit of detection of 46.8 nM, were tested in real water systems. In these experiments, the N/Al-CDs effectively measured the levels of Mn^7+^. This study introduces a novel and streamlined approach for the synthesis of a cellulose-based fluorescent nanoprobe, specifically designed for the detection of Mn^7+^ ions in environmental water systems [61].

#### 4.1.3. Detection of Hg(II)

Confirming the sensitivity of Au-N-CQDs for the detection of Hg(II) ions involved the addition of a 60 μL aliquot of Au-N-CQDs solutions (diluted to a 1/20 ratio) to a 2 mL volume of deionized water by adding 10 μL of Hg(II) (1 mM) into the solutions. The Hg(II) ions concentration was 0, 4.83, 9.615, 19.05, 28.30, 37.38, 41.86, 46.30, 55.05, 63.6, and 75.47 μM, respectively. The fluorescence intensity of Au-N-CQDs was gradually decreased when exposed to varying concentrations of Hg(II) at 360 nm.

Adding Hg^2+^ into the Au-N-CQDs solution resulted in a significant fluorescence quenching, as observed in Figure 7b. In contrast, the other metal ions lead to a minor quenching effect on fluorescence, which can be considered negligible. The evidence demonstrates the high selectivity of the Au/N-GQDs fluorescence sensor in detecting Hg(II), with a detection limit of 0.118 μM and a linear range from 0 to 41.86 μM. The identification of Hg(II) ions, thereby demonstrating a wide range of potential applications in the environmental and biological realms. The fluorescence of Au/N-GQDs containing Hg(II) in the “off” state was activated by the addition of EDTA-2Na or I^−^. This resulted in a fluorescence intensity recovery of 90% compared to the blank sample. Most importantly, the utilization of Au/N-CQDs proved to be effective in the precise identification of Hg(II) within a tap water sample. The current strategy may provide novel perspectives and methods for detection [44].

#### 4.1.4. Detection of Hg(II)

The study aimed to investigate the sensitivity and selectivity of Mg-N-CQDs in detecting Hg(II) by measuring the fluorescence intensity of the system upon adding various concentrations of Hg(II). The findings indicate that Mg-N-CQDs exhibited a gradual decrease in fluorescence intensity as the Hg(II) concentration increased (as illustrated in Figure 7a) [48]. This phenomenon can be explained based on the electron-transfer process from the surface of excited Mg-N-CQDs to the d-orbital of Hg(II), as previously reported in the literature [62,63,64]. An approximate detection limit for Hg(II) was determined to be 0.02 µM using the 3/slope method. In order to assess the selectivity of the system, an additional fifteen metal ions, namely, Na(I), K(I), Ag(I), Mg(II), Ca(II), Al(III), Fe(II), Co(II), Cu(II), Cd(II), Mn(II), Fe(III), Pb(II), Ni(II), and Zn(II), were selected to evaluate the impact of system on the fluorescence intensity. Although the concentration of Hg(II) ions was 5 µM and that of other metal ions was 15 µM, Hg(II) exhibited the most potent capacity to attenuate the fluorescence of Mg-N-CQDs. The observed phenomenon can be ascribed to the interplay between Mg-N-CQDs and Hg(II), producing a complex that lacks fluorescence. In contrast to other metal ions, the fluorescence of the system exhibits minimal impact, thereby demonstrating the high selectivity of the system employed for Hg(II) detection [65].

#### 4.1.5. Selective and Sensitive Detection of Cu^2+^

Comparative experiments were conducted to assess the sensitivity and selectivity of Fe-N-CQDs towards Cu^2+^ ions in the sample. The metal cations and anions including K^+^, Mg^2+^, Ca^2+^, Cu^2+^, Co^2+^, Zn^2+^, Ba^2+^, Ni^2+^, Fe^3+^, Pb^2+^, Cd^2+^, and Cr^3+^, as well as Cl^−^, Br^−^, HCO_3_^−^, CO_3_^2−^, NO_3_^−^, and SO_4_^2−^, were tested [66]. Given the marginal impact of reduced metal ion concentrations on fluorescence quenching and the typically low levels of metal ion concentrations in real-time scenarios, a metal ion selectivity experiment was conducted with a concentration of 1 mmol L^−1^ to ensure experimental precision [67]. The findings validate the exceptional specificity of Fe-N-CQDs towards Cu^2+^. In addition, Fe-N-CQDs’ sensitivity towards Cu^2+^ was assessed by measuring the emission fluorescence intensity of Fe-N-CQDs at varying concentrations of Cu^2+^ (ranging from 0–1000 mmolL^−1^). A gradual decrease was observed in the emission peak of Fe-N-CQDs at 340 nm with an increase in Cu^2+^ concentration, indicating high sensitivity towards Cu^2+^. A substantial linear range from 100 to 1000 nmol L^−1^ (with an R-squared value of 0.997). The Fe-N-CQDs demonstrate a calculated detection limit of 59 nmolL^−1^. The findings suggested that Fe-N-CQDs exhibit high sensitivity towards Cu^2+^ detection and hold significant promise for its detection applications [68,69,70,71,72].

#### 4.1.6. Discrimination between the Isomers of Phenylenediamine

According to prior reports, OPD, PPD, and MPD oxidized products were in different colors in the presence of a catalyst and H_2_O_2_ [73,74,75]. The results depicted in Figure 8a indicate that OPD and PPD oxidized with H_2_O_2_, forming yellow and brown products. The maximum absorption for the characterization of compounds was observed at 413 nm and 500 nm, respectively. In contrast, no apparent variation in color or distinct absorption peak was observed due to the oxidized product of MPD, as the pale yellow–green color appeared with N,Cu-CD. The findings recommended that PPD and OPD exhibit greater oxidation susceptibility than MPD.

According to the results, no effect was observed in absorption or color change. Based on the facts, the N-Cu-CD has been utilized to visually differentiate OPD, MPD, and PPD. However, it was noted that MPD did not exhibit any visible coloration under operational circumstances; thus, it was not considered for optimized investigations. The study was designed to assess the impact of various factors, including pH value, temperature, H_2_O_2_ concentration, and reaction time, on the quantitative analysis of OPD and PPD. The ultimate goal was to attain the most favorable analytical performances for detecting these compounds. The most favorable conditions for the experiment were determined to be a pH of 6.5 at 35 °C with a 15 mM concentration of hydrogen peroxide and a reaction time of 40 min. Regrettably, the reusability of N,Cu-CD is limited due to their diminutive size, rendering them unsuitable for recycling purposes. This study was conducted to discriminate between isomers of phenylenediamine by utilizing the catalytic properties of N,Cu-CDs in the oxidation process mediated by H_2_O_2_. This discrimination was achieved by observing the distinct colors exhibited by the oxidation products of OPD, MPD, and PPD. The colorimetric method demonstrates exceptional discriminatory capabilities, as well as favorable levels of selectivity and sensitivity. The technique employed in this study has yielded satisfactory outcomes in the identification of OPD and PPD in samples of natural water. These findings indicate the potential of this method for use in environmental monitoring [31].

#### 4.1.7. Estimation of Pyrogallol

1.2 mL stock solution of Cu-N@CDs (with a final concentration of 0.4 mgmL^−1^) was prepared by mixing with 0.5 mL Britton Robinson buffer at a pH of 8.4. The desired amount of PGL sample solution was added to the mixture, ranging from 0.15 to 70 μM for fluorimetric analysis and 6 to 140 μM for colorimetric analysis. The resulting mixture was thoroughly blended and diluted to a final volume of 3 mL using deionized water. Fluorescence spectra were measured at 370 nm, while the absorbance spectra were at 345 nm after a 20 m reaction time. The fluorescence intensity of Cu-N@CDs reduced gradually as the PGL concentration increased [32]. Efficiency in fluorescence quenching increased in a linear relationship with the concentration of PGL with a range of 0.15–70 μM. The calibration graph is represented by equation I/I_0_ = (1 + K_SV_).

Whereas “Io” is fluorescence intensity in the absence of PGL, “I” is fluorescence intensity in the presence of PGL, and K_SV_ is the static Stern–Volmer constant, which measures the conditional stability constant. The absorption spectra for measuring the PGL solution in the visible region, as depicted in Figure 8c,d exhibit similarities to that of Cu-N@CDs, which was observed at 345 nm. The obtained outcome indicates that the complexation reaction between PGL and Cu in the N@C-dots system yields a product with distinct yellow coloration.

The selectivity of Cu-N@CDs for PGL was evaluated by measuring the fluorescence in the presence of various ions using identical experiments. Specific cation solutions subjected to testing exhibit a degree of turbidity that does not extend to the point of precipitation, thus preventing the need for additional treatment measures. Additionally, an examination was conducted on the impact of several associated compounds, specifically catechol, phenol, resorcinol, hydroquinone, 1-naphthol, glycerol, and 2-naphthol. The study did not observe any significant interference. Nevertheless, gallic acid responded similarly to the mentioned PGL, likely attributed to the PGL functional group within its structure. The proposed mechanism is closely associated with the PGL group, thus indicating selectivity towards this moiety [76].

### 4.2. Applications of Mixed-Doped CQDs in Bio-Sensing

#### 4.2.1. Investigation of Amoxicillin

The study examined the effectiveness of Cu, N@CQDs when used with AMX, along with 4-NH_2_-APE and aqueous H_2_O_2_ solution. In previous studies, the compound 4-NH_2_-APE has been employed to quantify AMX [77]. The highly colored pink quinoneimine product was obtained through the oxidative coupling of AMX and 4-NH_2_-APE in an alkaline medium (pH 10.0) with H_2_O_2_. This product was subsequently used for the colorimetric quantification of AMX at a wavelength of maximum absorption (λmax) of 505 nm. The absorption spectra within the 300–650 nm range were obtained during the interaction between AMX and 4-NH_2_-APE in the presence of H_2_O_2_ and an alkaline medium with a pH of 10. An absorption band at 505 nm was observed in the presence of Cu,N@CQDs, hydrogen peroxide (H_2_O_2_), 4-aminothiophenol (4-NH_2_-APE), and 4-aminomethylbenzoic acid (AMX). The experiments involved the following systems: Cu, N@CQDs + H_2_O_2_ + 4-NH_2_-APE, Cu, N@CQDs + 4-NH_2_ APE + AMX, and Cu, N@CQDs + H_2_O_2_ + AMX. Figure 8c illustrates the process of quenching Cu, N@CQDs by the H_2_O_2_, 4-NH_2_-APE, and AMX systems. Implementing this strategy has proven an effective approach to identifying and detecting AMX.

In contrast, the fluorescence emission of Cu, N@CQDs did not exhibit any discernible alteration when subjected to the other systems. The observed decrease in fluorescence intensity of Cu, N@CQDs can be explained by the overlap between the absorption band of the resulting chromogen at 505 nm. The quenching phenomenon suggested the result of the inner-filter effect (IFE) [77,78]. Moreover, the observed fluorescence quenching of Cu, N@CQDs can be ascribed to quenching mechanisms. Static quenching typically occurs without any alteration in the fluorescence lifetime, whereas dynamic quenching is associated with changes in the fluorescence lifetime [79]. The confirmation of the extent of interaction between the chromogen and Cu, N@CQDs through static quenching was additionally established using the Stern–Volmer equation [80]. The study confirmed that the observed quenching phenomenon can be attributed to static quenching [81]. It can be inferred that the interaction between the chromogen and Cu, N@CQDs results from this static quenching mechanism.

#### 4.2.2. Determination of Uric Acid (UA)

In this experiment, a 400 μL volume of uricase solution containing a concentration of 2.5 µmL^−1^ was mixed with a uric acid (UA) standard solution, which had a concentration of 0.1 mM. This combination was done in a 1.0 mL solution of boric acid–borax buffer. Then, the mixture was exposed to an incubation period lasting 15 min at 35 °C. Subsequently, a HAc/NaAc buffer solution with a concentration of 0.2 M and a pH of 3.0, totaling a volume of 2.0 mL, was successively mixed with 100 μL of Fe@NCDs and 200 μL of TMB, the latter being dissolved in an ethanol solution. The solution was agitated and then transferred to a water bath maintained at a constant temperature of 45 °C for 50 min to promote a sufficient reaction. The UV-visible spectrophotometry method was utilized to quantify the absorbance of the resulting solution at a specific wavelength of 652 nm [82]. The current research examines the colorimetric detection of standard solutions of uric acid (UA) through the catalytic activity of peroxidase-like Fe@NCDs, under conditions that have been optimized. The UV-vis spectrum showed a progressive rise in absorbance at 652 nm with increasing concentrations of UA in the solutions. The linear correlation between the absorbance at a wavelength of 652 nm and the concentration of UA was observed in the range of 2–150 μM.

In order to apply uric acid to real samples, three urine samples from human subjects were acquired from the Second Nanning People’s Hospital. To eliminate most impurities, the collected urine samples underwent centrifugation at 22,000 revolutions per minute for 15 min. Considering the established range of 1.49 to 4.46 mM for the typical urinary uric acid (UA) levels in humans, a 100-fold dilution was implemented on the sample. This dilution was performed to ensure that the concentrations of UA fell within the linear range, thereby minimizing the potential interference from other components present in urine samples. Following the pretreatment procedure, the diluted urine underwent additional processing as outlined previously, and the resulting signals were collected at a wavelength of 652 nm [30,83,84,85,86,87,88].

#### 4.2.3. Detection of Tetracyclines (TCs)

Tetracyclines, a class of antibiotics commonly employed in animal husbandry, are used for infection prevention and treatment. Therefore, detecting the residues of TCs in meat and milk products is crucial to guarantee the food’s safety [89]. The present study assessed the viability of the analytical approach described by employing the standard addition method to detect residues of TCs in real-life samples by Ce-N-CQDs [90]. The present study involved the computation of the regaining rate of milk and pork specimens blended with varying quantities of DC, DC + TC, and CTC + DC. The recovery percentage of typical specimens exhibited a range of 92% to 107.06%, accompanied by a consistency of 1.05% to 2.84% (RSD%, n = 3). The findings confirm that Ce-N-CQDs exhibit high precision, minimal interference from external factors, and hold significant potential for deployment in food surveillance.

The compound known as DC shows a broad absorption peak at wavelengths of 278 nm and 350 nm. In contrast, Ce–N–CDs display a broad absorption peak at 348 nm, which coincides with its excitation peak of approximately 348 nm. This excitation leads to light emission at a wavelength of 440 nm. Consequently, the considerable similarity between the absorption spectra of doxycycline and Ce-N-CDs may result in a prominent internal filtering effect (IFE) [91,92], resulting in a notable reduction in the fluorescence signal of Ce-N-CDs at 440 nm. Simultaneously, the fluorescence lifetime of Ce-N-CDs was determined by fitting the fluorescence lifetime in the absence and presence of DC. There was no observed alteration in the fluorescence lifetime of Ce-N-CDs. Hence, based on the aforementioned findings, it can be inferred that the observed fluorescence quenching of Ce-N-CDs caused by DC can be ascribed to the internal filtering effect (IFE). It provides ideas for synthesizing nanomaterials from waste biomass for peroxidase activity [40,93].

#### 4.2.4. Determination of Oxytetracycline (OTC)

Milk was chosen for precise sample analysis to enhance the credibility of the established Ce-N-CQDs@ZIF-67@MIP FL sensing assay. The spiking and recovery trials were conducted by analyzing milk samples spiked with varying concentrations of oxytetracycline (OTC), specifically, 0, 1, 5, and 15 μgmL^−1^. The results of the FL sensing method utilizing Ce, N-CDs@ZIF-67@MIP demonstrated a satisfactory recovery rate ranging from 96.05% to 99.64%. The findings indicate a strong correlation between the results obtained through the constructed FL approach and those determined by traditional HPLC [94]. These results confirm the accuracy and reliability of the FL approach in quantitatively determining OTC in dairy products. These findings suggest that the sensor is a viable option for detecting OTC in real-world samples.

The selectivity of the developed fluorescence sensor was evaluated by choosing compounds (TET, DOX, CAP, and TOB) that have comparable structures to the target OTC as potential interfering substances. The fluorescence (FL) response of Ce, N-CDs@ZIF-67@MIP to OTC was significantly more significant compared to other analytes. This enhanced response can be attributed to the presence of recognition sites within Ce, N-CDs@ZIF-67@MIP that closely match the size, shape, and functional groups of OTC. Consequently, the prepared probe exhibited an exceptional capacity for recognizing OTC. However, the fluorescence (FL) response of Ce, N-CDs@ZIF-67@NIP towards OTC was found to be similar to its response towards other analogs. This similarity can be attributed to the nonspecific physical adsorption between Ce, N-CDs@ZIF-67@NIP, and these substances. The aforementioned findings indicate that the developed Ce, N-CDs@ZIF-67@MIP probe exhibited remarkable selectivity and specificity towards OTC [41,95].

#### 4.2.5. Determination of Ofloxacin (OFL)

The origin of domestic water is typically local and does not undergo pretreatment procedures. Milk samples were collected from a nearby supermarket and subjected to pretreatment procedures. Briefly, a combination of milk (1 mL) and acetonitrile (4 mL) was introduced, followed by sonication of the mixtures for 15 min. The samples underwent centrifugation at 5000 revolutions per minute for 15 min to eliminate the denatured protein precipitate. The liquid portions were obtained and filtrated using a 0.22 μM organic filter membrane to eliminate protein residue. Milk samples for treatment were subjected to a 10-fold dilution using phosphate buffer with a pH of 7.4. Subsequently, varying concentrations of ofloxacin (0.08, 0.4, and 2 μM) were introduced into the solution. The OFL quantity was determined using a calibration curve and the fluorescence intensity of β-CD-N-Zn-CDs [36,96].

The fluorescence intensity of N-Zn-CDs was increased by using OFL; however, no red shift in fluorescence was observed. The N-Zn-CDs exhibit abundant carboxyl and hydroxyl functional groups. The benefits mentioned above afford a prospect for developing hydrogen bonds between OFL and N-Zn-CDs, thereby fostering the generation of augmented fluorophores and yielding a greater fluorescence quantum under identical excitation light. The introduction of Zn^2+^ ions has been observed to have a positive impact on the augmentation of hydrogen bonding between OFL and CDs. This, in turn, facilitates the production of fluorophores through the mechanism of fluorescence resonance energy transfer (FRET) [97].

#### 4.2.6. Sensing of Cholesterol

A mixture containing 200 µL of varying cholesterol concentrations and 10 µL of ChOX solution with a concentration of 0.6 mgmL^−1^ was prepared to detect cholesterol. This mixture was combined with 190 µL of OPD solution with a concentration of 14 mM (prepared in 10 mM PB, pH = 6.6) and 10 µL of HRP solution with a concentration of 1 mg mL^−1^ (prepared in 10 mM PB, pH = 6.6). The resulting solution was incubated at 37 °C without light for 30 min. A volume of 1 mL of the serum was combined with cholesterol esterase (5 L, 5 mg/mL) to detect cholesterol in human serum. It was diluted by a factor of 200 using phosphate buffer (PB). The measurement of the diluted serum was conducted following the procedure mentioned above. Under the conditions found to be most favorable (as shown in Figure 9b), a corresponding decrease in the fluorescence intensity of N-Cu-CDs at 460 nm was observed as the cholesterol concentrations increased up to 0.60 mM [33,98,99,100].

#### 4.2.7. Detection of Ascorbic Acid

The doping of Cu and N elements has been identified as a crucial factor in enhancing the detection capability of carbon dots. Introducing N and Cu elements results in a positive surface charge and increases the interaction of the analyte with cyclodextrins [101,102]. Additionally, the ability of copper to readily form chelates with a wide range of small molecules enhances the sensitivity and selectivity of cyclodextrins toward the target. Ascorbic acid (AA) reduced the emission spectra of Cu-N-CDs in a phosphate-buffered saline (PBS) solution with a pH of 7.4. Figure 9a demonstrates a gradual decrease in the fluorescence intensity of Cu-N-CDs at 470 nm by increasing the concentration of AA. The selectivity of Cu-N-CDs in the fluorescence response towards metal ions and other small organic molecules was inspected for AA. The Cu-N-CDs exhibited unique selectivity, with the most pronounced fluorescence quenching observed in the presence of AA.

Generally, two observed mechanisms for quenching fluorescence are dynamic and static. Dynamic quenching is the interaction between quenchers and a fluorescent material influenced by diffusion. Static quenching is the formation of the complex formed between the quencher and a fluorescent substance. Hence, the static quenching proved advantageous in investigating the surface state of carbon dots and subsequently explaining the fluorescence origin of CDs [34,103,104,105].

#### 4.2.8. Detection of Cholesterol and Uric Acid

A novel fluorescence probe was developed to detect uric acid and cholesterol. This probe exhibited a response to the enzyme-catalyzed generation of H_2_O_2_. Cholesterol and uric acid produce hydrogen peroxide (H_2_O_2_) during oxidation reactions in the presence of oxygen (O_2_), provided by their oxidoreductase enzymes. Subsequently, H_2_O_2_ catalyzes the oxidation of OPD (o-phenylenediamine) to yield DAP (diaminophenazine) with the assistance of HRP. The fluorescence of various carbon-based nanomaterials can be effectively quenched through intramolecular Förster resonance energy transfer (IFE) by utilizing a DAP with an emission wavelength of 540 nm [106]. Hence, these N-Co-CDs possessing exceptional characteristics can serve as highly effective ratio-metric fluorescent probes for indirectly identifying cholesterol and uric acid. The potential mechanism for fluorescence quenching in the N-Co-CDs/DAP system is likely to remain intramolecular Förster energy transfer (IFE) [55].

The detection of cholesterol in human blood serum was performed by mixing a 500 μL sample of human blood serum with 10 μL cholesterol esterase solution (1 mg/mL) to catalyze the hydrolysis of cholesterol ester, producing free cholesterol. Subsequently, the mixture was diluted 100 times by adding 10 mM phosphate buffer with 6.5 pH. To detect uric acid in human blood serum, a 100-fold dilution of the serum sample was achieved by adding 10 mM phosphate buffer with a pH of 6.5 (Figure 10b,c) [33].

#### 4.2.9. Detection of Hematin

The potential sensing capabilities of Fe-N-CDs were explored by selecting hematin as a test material. The detection of hematin was performed by dissolving Fe-N-CDs in deionized water at room temperature. Maximum excitation occurred at 365 nm; thus, sensing performance was evaluated at 365 nm. The fluorescence intensity of Fe-N-CDs exhibited a gradual decrease different concentrations of hematin were added. Evidently, upon introducing 27 μM hematin, the reduction in fluorescence intensity experienced approximately 82%. The findings of this study demonstrated that hematin exhibited a significant ability to suppress the fluorescence by Fe-N-CDs [29].

Nevertheless, there was no direct correlation between the decrease in fluorescence intensity and the amount of hematin added. As the level of fluorescence reduced, a greater quantity of hematin was required to attain the initial quenching phenomenon. The determined limit of detection (LOD) exhibited by the Fe-N-CDs was significantly lower than in recent studies for detecting hematin using carbon dots. This observation suggests that the Fe-N-CDs have remarkable sensitivity towards hematin [107,108].

### 4.3. Bio-Imaging

#### 4.3.1. Bio-Imaging of Zr-N-CDs

The current study aimed to examine the cytotoxic impacts of Zr-N-CDs on HeLa cells using the MTT assay. The purpose was to investigate and analyze the possible uses of Zr-N-CDs in bio-imaging and cellular imaging of chromium (Cr), as illustrated in Figure 11. No detectable toxicity was observed in the proliferation of HeLa cells when cultured in different concentrations (10, 20, 30, and 40 mL) of synthesized Zr-N CDs. At a constant concentration of 40 mL, the observed rate of proliferation of cells remained at approximately 85%, indicating that Zr-N-CDs demonstrate relatively minimal cytotoxic effects. Moreover, in practical scenarios, a markedly reduced concentration of Zr-N-CDs is required for bioimaging. Based on the abovementioned analysis, Zr-N-CDs are expected to be used for bioimaging applications [54].

The examination of heavy metal monitoring in the biological milieu and human organisms has attracted considerable interest owing to its correlation with cytotoxic effects, oxidative stress, and carcinogenic properties [109]. Zr-N-CDs were employed in a bio-imaging investigation of human cervical cancer cells (HeLa cells) due to their best biocompatibility, stability, low cytotoxicity, uniform dispersion, and small size. Figure 10 demonstrates that the fluorescence intensity of Zr-N-CDs in cells remained unchanged when exposed to Cr(III). The observed result can be ascribed to the complex process of cellular uptake of Cr(III), which subsequently increases the cell membrane’s permeability. Zr-N-CDs’ fluorescence was observed to undergo quenching in the presence of Cr(VI) within cells. This quenching phenomenon can be attributed to the effect of fluorescence resonance energy transfer (FRET). Hence, it can be inferred that Zr-N-CDs exhibit promising capabilities for cellular bioimaging of Cr(VI) [54].

#### 4.3.2. Cell Imaging

Confocal microscopy was employed to investigate the probable usage of Fe-N-CDs in the context of cellular imaging and the monitoring of Fe^3+^ at the cellular level. The Fe-N-CDs and LysoTracker red was differentiated using blue and red colors, as depicted in Figure 12. Figure 12A displays fluorescent images of HeLa cells treated with a concentration of 500 μgmL^−1^ Fe-N-CDs. The internalized pathway of the Fe-N-CDs was tracked using LysoTracker Red. Figure 12B,C depicted the increased overlap of blue and red fluorescence, indicating that most prepared CDs were localized within the lysosome. At the same time, a separate portion was distributed throughout the cytoplasm.

Quantitative analysis and examination of confocal microscopic images showed that HeLa cells marked with Fe-N-CDs, and LysoTracker Red DND-99, exhibited a substantial level. The findings suggested that the Fe-N-CDs demonstrated remarkable lysosomal targeting capabilities. Upon introducing exogenous Fe^3+^ into the HeLa cells previously treated with Fe-N-CDs, an apparent attenuation in the blue fluorescence emitted by the Fe-N-CDs was observed (refer to Figure 12D). Figure 12E,F depict the confocal microscopic red channel images in the merged form. The observed overlap in these images aligns with the findings presented in Figure 12C. As mentioned earlier, the observations suggest that the synthesized Fe-N-CDs with vibrant blue fluorescence possess promising prospects for detecting Fe^3+^ in cellular systems [29,107].

**Figure 12 nanomaterials-13-02336-f012:**
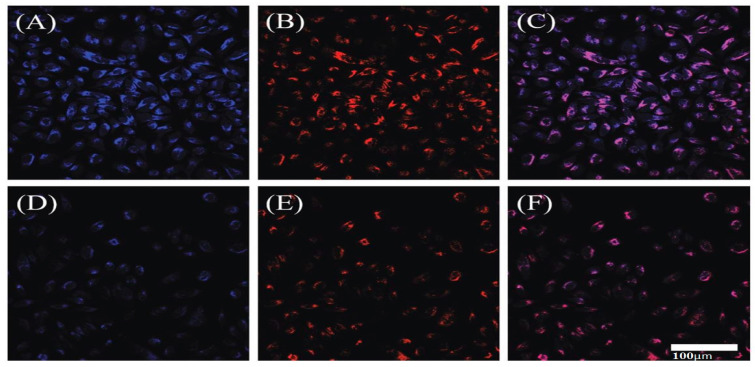
Illustration of cell imaging and monitoring of Fe(III). Reproduced from [29], with permission from RSC 2021.

#### 4.3.3. Applications in Cellular Imaging

Carbon dots (CDs) in bio-imaging comprise a notable application due to their exceptional fluorescence characteristics and strong compatibility with biological systems. The IC_50_ value was determined to be 812.96 μgmL^−1^, suggesting that the Cu-N-CDs demonstrated favorable biocompatibility and low cytotoxicity. The fluorescent images of HepG2 cells were acquired after incubation in a concentration of 100 μgmL^−1^ of Cu-N-CDs for 6 h at 37 °C. These images were obtained using various light excitations. The study’s findings indicate that utilizing Cu-N-CDs as labels for HepG2 cells enables multicolor cellular imaging [34,110].

Furthermore, the research revealed that Cu-N-CDs’ luminescence predominantly occurs within the cytoplasm rather than the cellular nucleus. This finding proves that Cu-N-CDs can cross the cellular membrane barriers and enter the intracellular region but cannot penetrate cellular nuclei [65,111,112,113].

#### 4.3.4. Investigation of the Bacteriostatic Effect of Zn-N-CQDs on Gram-Negative *E. coli* Cells

Zn-N-CDs exhibit interesting bacteriostatic properties by inhibiting the proliferation and replication of bacteria. A volume of 100 μL of sterile Luria–Bertani (LB) medium was combined with an equal volume of zinc-nitrogen-carbon-dots (Zn-N-CDs) of a predetermined concentration. The mixture was injected with Escherichia coli DH5α cells and subjected to overnight incubation at 37 °C. The observed growth inhibition displays a descending trend, with Zn-N-CDs6 (2 mg/mL) exhibiting the highest inhibition, followed by Zn-N-CDs5 (1 mg/mL), Zn-N-CDs4 (0.5 mg/mL), Zn-N-CDs3 (0.25 mg/mL), Zn-N-CDs2 (0.125 mg/mL), and Zn-N-CDs1 (0.062 mg/mL), respectively. The present study employed a positive control of LB media with bacterial inoculum without incorporating Zn-N-CDs.

A negative control was implemented, consisting solely of media without any inoculum to confirm the absence of any potential contamination. Based on the concentrations present in the microtitre plate, it was determined that no real growth was observed at a concentration of 1 mg/mL. The study determined that the minimum inhibitory concentration (MIC) of Zn-N-CDs was 1 mg/mL.

Upon elevating the concentration range (0–2 mg/mL) of Zn-N-CDs, a corresponding escalation in the growth inhibition of *E. coli* was observed. The findings suggest that the concentration of N, Zn-CDs significantly influenced bacteriostatic activity. The present study investigated gram-negative bacteria Escherichia coli growth kinetics in Luria–Bertani broth supplemented with synthesized zinc–nitrogen–cyclodextrin complexes under various experimental conditions. The study revealed that Zn-N-CDs synthesized in a zero-dimensional form and with varying concentrations exhibit inhibitory effects on the proliferation of *E. coli*. The bacteriostatic effects were investigated over time by assessing optical density values at a wavelength of 600 nm (OD600) in the bacterial culture, which had been enriched with varying concentrations of Zn-N-CDs. Three sets of 100 mL sterile LB media were prepared, each with varying concentrations of Zn-N-CDs (control, 0.5 mg/mL, and 1 mg/mL). Then, these media were inoculated with *E. coli* DH5α cells from an overnight culture to achieve an initial OD value of 0.05. The sample was incubated under controlled temperature and agitation conditions, specifically at 37 °C and 200 rpm. The optical density of the sample was subsequently determined at various time points by measuring its absorbance at a wavelength of 600 nm.

The experimental results indicate that using 0.5 mg/mL Zn-N-CDs in the media led to partial inhibition of bacterial growth. At a concentration of 1 mg/mL, Zn-N-CDs exhibited significant inhibitory effects on the proliferation of *E. coli*. Without Zn doping, treatment of *E. coli* cells with control CDs at a concentration of 1 mg/mL did not inhibit growth. A significant reduction in bacterial colony formation was observed upon exposure to 1 mg/mL of Zn-N-CDs. Conversely, treating bacterial cells with identical concentrations of control CDs did not elicit any discernible impact on cell viability [114].

#### 4.3.5. Cell Labeling and Cytotoxicity

The cytotoxicity of the as-prepared Mg-N-CDs was evaluated using the MTT assay method before their application. The cytotoxicity of the CDs was found to be low, as evidenced by the viability of over 90% of cells that were incubated in a medium containing CDs at a concentration of 250 mgmL^−1^ or lower. An exploratory experiment was conducted to evaluate the potential utility of CDs in the context of cell imaging. The L929 cells were subjected to a culture process in a medium that contained 100 mgmL^−1^ CDs for 24 h. Following this, the cells were washed thrice with PBS and, subsequently, fixed with a 4% paraformaldehyde solution in PBS at a temperature of 4 °C overnight. Then, the cells were subjected to observation under a laser-scanning confocal microscope. The results indicate that the L929 cells labeled with CDs exhibited significant fluorescence upon excitation at 405 nm, 488 nm, and 543 nm.

Conversely, the control cells displayed minimal fluorescence under identical conditions. The specific emission characteristics of Mg-N-CDs, along with the fluorescence microscopy images of droplets containing CDs under various light excitations, including bright field, ultraviolet, blue, and green light, suggest that the labeled L929 cells with CDs would exhibit multicolor fluorescence when excited at different wavelengths. This gives us a wider range of options to observe samples labeled with CDs. Additionally, the distribution of CDs was primarily observed across the cell membrane and cytoplasmic region, with minimal photoluminescence detected in the majority of cell nuclei, suggesting a low concentration of CDs in these areas. This finding aligns with prior literature indicating that CDs can mark both the cellular membrane and cytoplasm yet exhibit challenges in penetrating the nucleus. In conjunction with the absence of blinking, which serves to reaffirm their high photostability, and the biocompatibility mentioned above, Mg-N-CDs have demonstrated significant promise in the realm of biomedical applications [47].

## 5. Conclusions

The synthesis of fluorescent CQDs through simultaneous doping with metals and non-metals has unlocked a world of possibilities in chemical sensing, biosensing, and bio-imaging. The exceptional optical properties of mixed-doped CQDs have paved the way for highly sensitive and accurate chemical detection. These nanomaterials hold immense potential in environmental monitoring, and biological and medical investigations. Mixed-doped CQDs have gracefully integrated themselves into the field of biology, offering real-time insights into biological processes. With their biocompatibility and non-invasiveness, they act as vigilant materials, monitoring biomolecules in living systems. This study opens new paths for disease diagnosis, drug development, and personalized medicine.

As the prospects of mixed-doped CQDs, diagnostics are more precise, therapies are more targeted, and our understanding of the complex workings of life is expanded. From revolutionizing healthcare to advancing scientific research, these nanomaterials hold immense promise. Improved synthesis methods enhance the stability and fluorescence of CQDs, enabling more precise imaging and diagnostics. Imagine personalized medicine where drug-loaded CQDs target specific cancer cells with pinpoint accuracy, minimizing side effects, and will serve as powerful tools for targeted drug delivery. We can guide these nanoparticles to particular cells or tissues in the body through precise surface modifications and functionalization, delivering outstanding therapeutic species with accuracy. Mixed-doped CQDs seamlessly integrate with surface modification with a polypeptide or any targeted ligand to make it beneficial for the diagnosis and targeted drug delivery.

## Figures and Tables

**Figure 1 nanomaterials-13-02336-f001:**
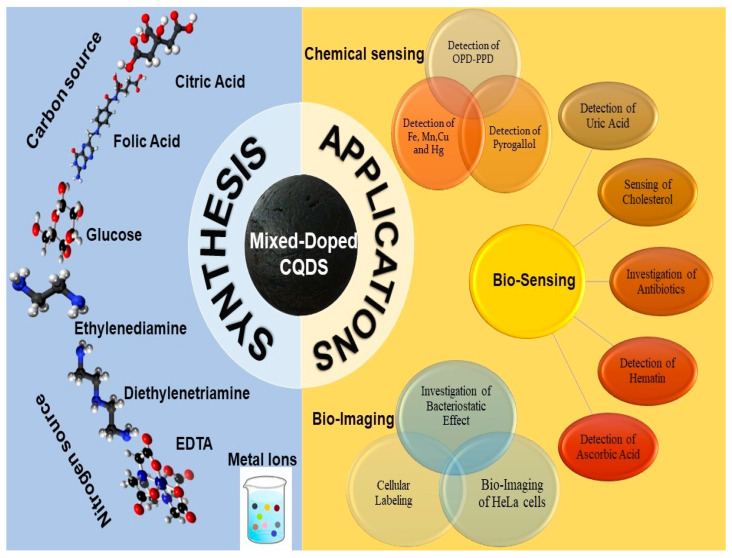
Illustration of the synthesis and applications of mixed-doped carbon quantum dots.

**Figure 4 nanomaterials-13-02336-f004:**
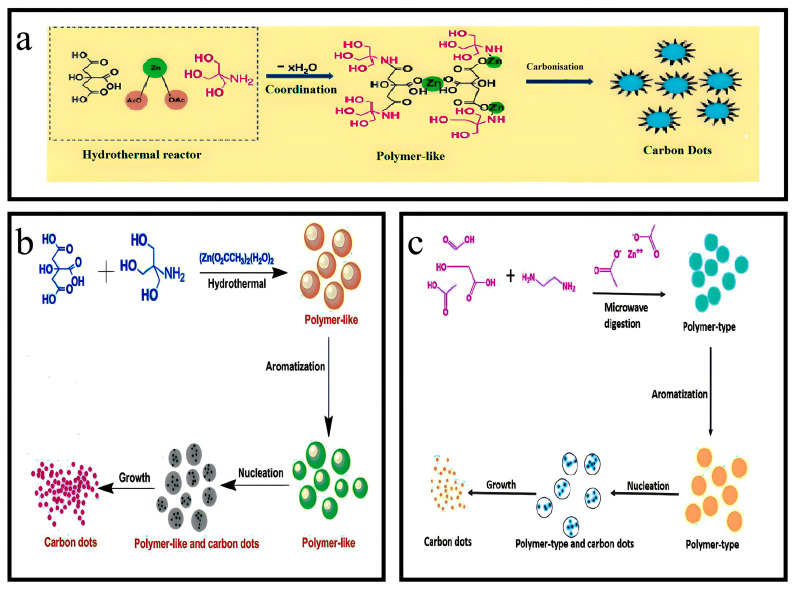
Overview of synthesis of zinc and nitrogen co-doped CQDs. (**a**) Synthesis of Zn-N-CQDs. Reproduced from [36,37], with permission from Elsevier 2020. (**b**,**c**) Synthesis of N,Zn-CDs. Reproduced from [38], with permission from Elsevier 2019.

**Figure 5 nanomaterials-13-02336-f005:**
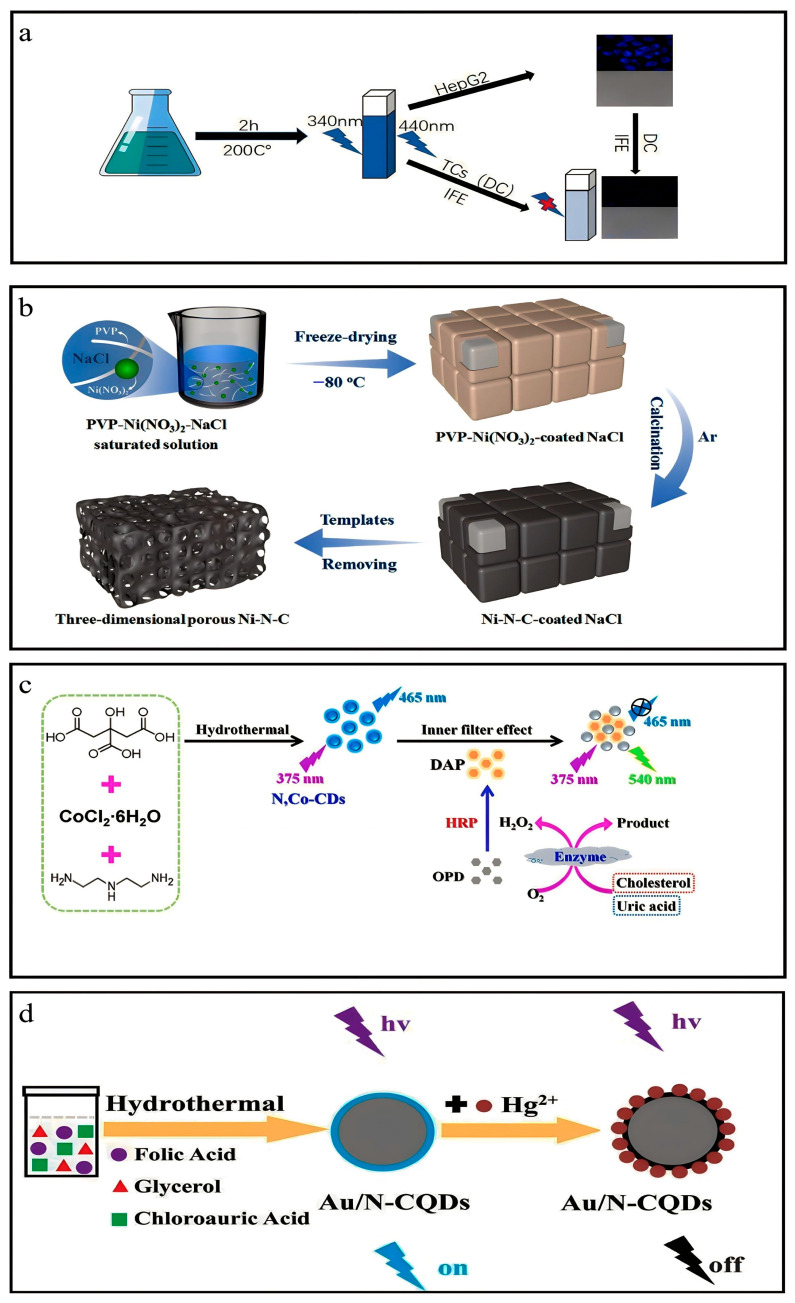
Overview of synthesis. (**a**) Synthesis of Ce-N-CQDs. Reproduced from [40], with permission from Elsevier 2021. (**b**) Synthesis of Ni-N-C materials. Reproduced from [42], with permission from Elsevier 2019. (**c**) Synthesis of N,Co-CDs. Reproduced from [43], with permission from ACS 2019. (**d**) Synthesis of Au/N-CQDs. Reproduced from [44], with permission from Elsevier 2018.

**Figure 6 nanomaterials-13-02336-f006:**
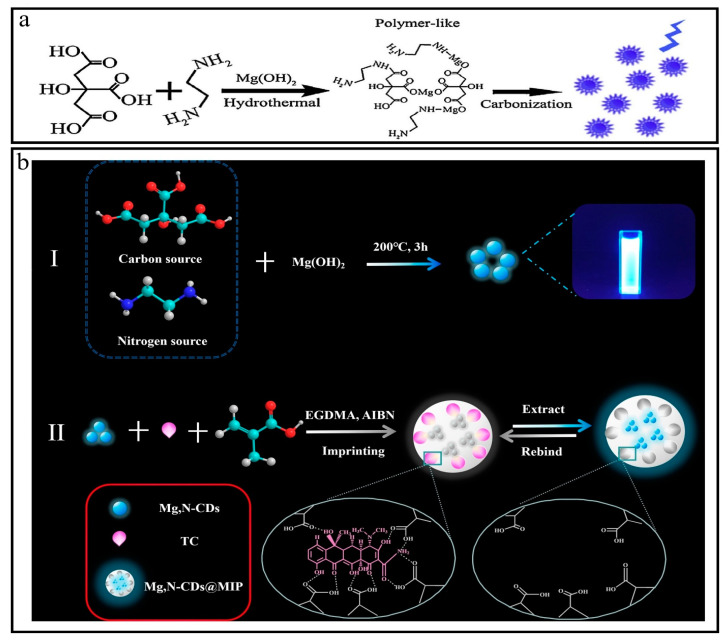
(**a**,**b**) Synthesis of Mg-N-CDs. Reproduced from [47,48,49], with permission from RSC 2014, Elsevier 2016.

**Figure 7 nanomaterials-13-02336-f007:**
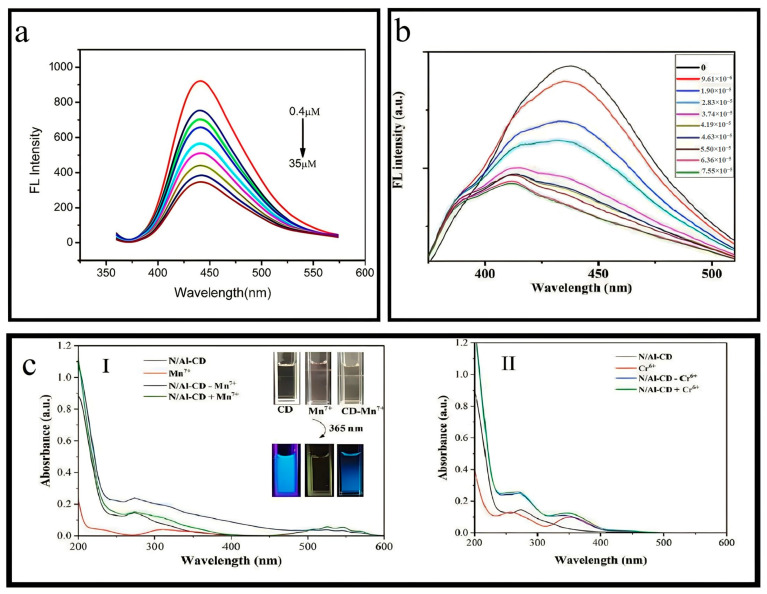
(**a**,**b**) Detection of Hg(II). Reproduced from [44,48], with permission from Elsevier 2016 and 2018. (**c**) Determination of Mn(VII). Reproduced from [53], with permission from Springer 2019.

**Figure 8 nanomaterials-13-02336-f008:**
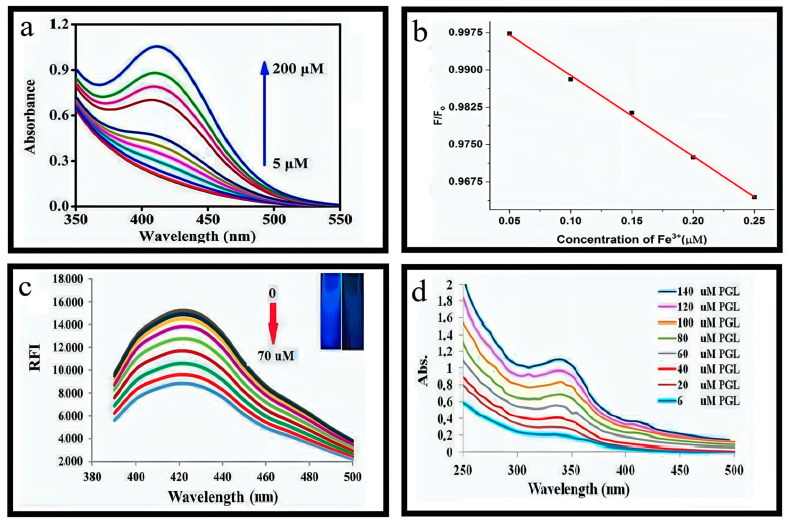
(**a**) Discrimination between OPD and PPD. Reproduced from [31], with permission from Springer 2019. (**b**) Detection of Fe^3+^. Reproduced from [38], with permission from Elsevier 2019. (**c**,**d**) Sensing of Pyrogallol. Reproduced from [32], with permission from Springer 2019.

**Figure 9 nanomaterials-13-02336-f009:**
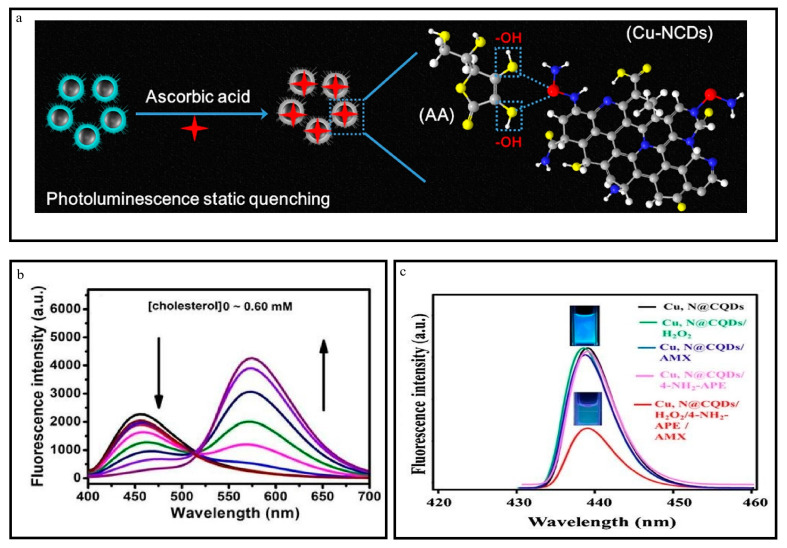
(**a**) Detection of Ascorbic acid. Reproduced from [34], with permission from Elsevier 2020. (**b**) Detection of cholesterol. Reproduced from [33], with permission from ACS 2017. (**c**) Determination of amoxicilin. Reproduced from [77], with permission from IOPscience 2019.

**Figure 10 nanomaterials-13-02336-f010:**
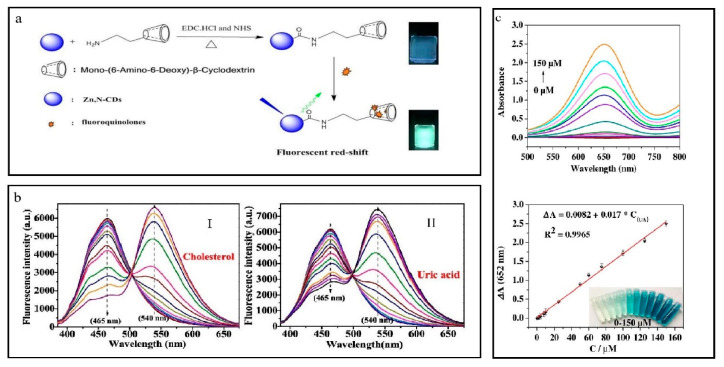
(**a**) Determination of OFL. Reproduced from [36], with permission from Elsevier 2020. (**b**,**c**) Detection of cholesterol and uric acid. Reproduced from [33], with permission from ACS 2017.

**Figure 11 nanomaterials-13-02336-f011:**
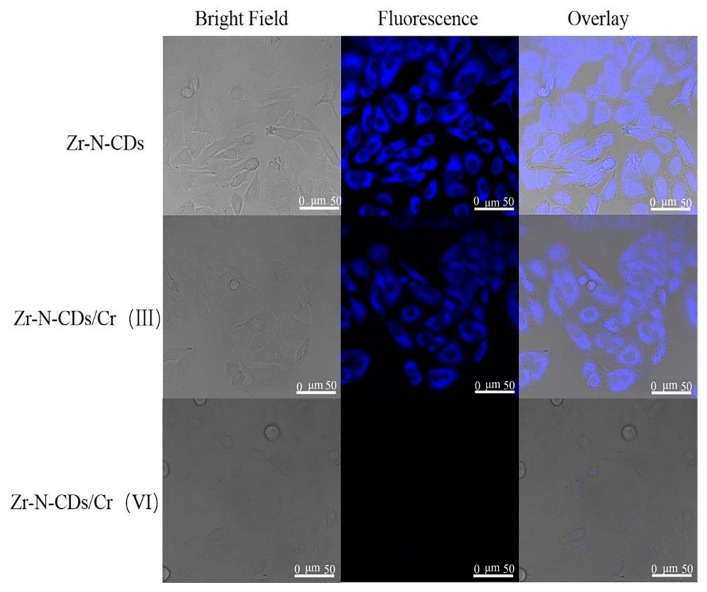
Illustration of bio-imaging of Zr-N-CDs. Reproduced from [54], with permission from Elsevier 2021.

## Data Availability

Data presented in this manuscript are available from the corresponding author upon reasonable request.

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
