# Peer review of "Recent Advancements in Metal and Non-Metal Mixed-Doped Carbon Quantum Dots: Synthesis and Emerging Potential Applications"

_nanomaterials, 2023, doi:10.3390/nano13162336_

Round 1
Reviewer 1 Report
The author describes the “Recent Advancements in Metal and Non-Metal mixed-doped Carbon Quantum Dots: Synthesis and Emerging Potential Applications”. This review article is quite interesting from a technological point of view. The author should revise their manuscript based on the comments and suggestions. I recommended a Minor revision of the manuscript.
The Minor suggestion below:
- In the abstract, the author can state a clear research question to convey the main objective of this review.
- The should provide appropriate keywords in the revised manuscript.
- The figure quality is too poor and the author should improve the quality of the figure.
- The author should include more application details of CQD in the revised manuscript.
- The authors should thoroughly review the manuscript for spelling and grammar errors. Numerous mistakes have been identified, and it is important to address them diligently.
- The author should add the recent relevant references in the revised manuscript.
Minor editing of English language required
Author Response
The given suggestions have been addressed in the response to reviewers report.

Reviewer 2 Report
In this review article the authors describe the synthesis and applications of carbon quantum dots, specifically focusing on doped carbon quantum dots. The review seems to be well-written and comprehensive in its description of the current field of the research. The beginning sections describing synthetic parameters for carbon quantum dots that have been reported seems tedious and unnecessary for a review article, but there is nothing explicitly wrong with it. This is a fine review article, and I have no outstanding concerns with its publication.
Author Response
Thank you so much for your appreciation.

Reviewer 3 Report
The authors selected for this paper the novelties in the development of CQD systems in different variants, briefly presenting the synthesis methods and potential applications.
A major advantage of the topic addressed is the fact that this review can be of real help in identifying a suitable solution for a particular study related to the information presented in the paper.
I agree with the acceptance of the work with only one amendment related to the figures present in the work, namely:
In some figures, resizing leads to the aesthetic alteration of the respective image, such as in figure 4a and figure 5. It is good to keep the original ratio.
Author Response
The given suggestions have been addressed accordingly.
